# ST-segment elevation myocardial infarction with non-obstructive coronary arteries: Score derivation for prediction based on a large national registry

**Magdalena Jędrychowska[1], Zbigniew Siudak[2], Krzysztof Piotr Malinowski[3], Łukasz Zandecki[2], Michał Zabojszcz[2], Tomasz Kameczura[4], Piotr Mika[5], Krzysztof Bartuś[6], Wojciech Wańha[7], Wojciech Wojakowski[7], Jacek Legutko[8], Stanisław Bartuś[1,3], Rafał Januszek[1,5]***

1 2nd Department of Cardiology and Cardiovascular Interventions, University Hospital, Kraków, Poland, 2 Collegium Medicum, Jan Kochanowski University, Kielce, Poland, 3 2nd Department of Cardiology, Jagiellonian University Medical College, Kraków, Poland, 4 Chair of Electroradiology, Faculty of Medicine, University of Rzeszów, Rzeszów, Poland, 5 Department of Clinical Rehabilitation, University School of Physical Education, Kraków, Poland, 6 Department of Cardiovascular Surgery and Transplantology, Jagiellonian University Medical College, John Paul II Hospital, Kraków, Poland, 7 Department of Cardiology and Structural Heart Diseases, Medical University of Silesia, Katowice, Poland, 8 Department of Interventional Cardiology, Institute of Cardiology, Jagiellonian University Medical College, John Paul II Hospital, Kraków, Poland

* jaanraf@interia.pl

**Data Availability Statement:** Data are available from the Open Science Framework (DOI: 10.17605/OSF.IO/MD7AT).

## Abstract

### Background

Acute myocardial infarction with ST-segment elevation (STEMI) and obstructive coronary arteries (MI-CAD) are treated with primary percutaneous coronary interventions (pPCI), while patients with STEMI and non-obstructive coronary arteries (MINOCA), usually require non-invasive therapy. The aim of the study is to design a score for predicting suspected MINOCA among an overall group of STEMI patients.

### Materials and methods

Based on the Polish national registry of PCIs, we evaluated patients between 2014 and 2019, and selected 526,490 subjects treated with PCI and 650,728 treated using only coronary angiography. These subjects were chosen out of 1,177,218 patients who underwent coronary angiography. Then, we selected 124,663 individuals treated with pPCI due to STEMI and 5,695 patients with STEMI and MINOCA. The score for suspected MINOCA was created using the regression model, while the coefficients calculated for the final model were used to construct a predictive model in the form of a nomogram.

### Results

Patients with MINOCA differ significantly from those in the MI-CAD group; they were significantly younger, less often males and demonstrated smaller burden of concomitant diseases. The model allowed to show that patients who scored more than 600 points had a

**Funding:** Publication financed by the Ministry of Science and Higher Education in the form of a grant awarded to RJ (022 / RID/ 2018/19). This was part of a program under the Ministry of Science and Higher Education called 'Regional Initiative of Excellence' in 2019-2022.

**Competing interests:** The authors have declared that no competing interests exist.

19% probability of MINOCA, while for those scoring more than 650 points, the likelihood was 71%. The other end of the MINOCA probability scale was marginal for patients who scored less than 500 points (< .2%).

## Conclusions

Based on the created MINOCA score presented in the current publication, we are able to distinguish MINOCA from MI-CAD patients in the STEMI group.

## Introduction

Acute myocardial infarction (AMI), with no relevant stenosis of the coronary artery, has been defined as MI with non-obstructive coronary arteries (MINOCA). Its frequency varies and ranges from a few percent to a dozen or so depending on the analysed cohort [1–4]. The mechanism of myocardium damage in patients with MINOCA is extremely diverse. It is associated with very multifarious etiology and diagnoses having exact reflection on prognoses and different treatments strategies. This disease affects both the large epicardial arteries and coronary microcirculation. Illnesses mimicking MINOCA and not directly related to the coronary arteries include, among others, myocarditis, pericarditis, prothrombotic conditions and ailments, stroke, sepsis, pulmonary embolism, kidney and heart failure, as well as other heart and large vessel diseases, causing malfunction, such as tachyaarhythmia, heart valve defects (aortic stenosis), cardiomyopathy (including stress-cardiomypathy) or aortic diseases [5,6]. The prognosis of MINOCA patients is independent of revascularisation. Therefore, in this group of patients, the improvement of health, quality of life, and thus—prognosis, depends on appropriate therapy, with regard to the etiology and MINOCA mechanism. Among AMI patients, those with ST-segment elevation myocardial infarction (STEMI) and relevant coronary atherosclerosis (MI-CAD) are of key concern because they require immediate coronary angiography and, possibly, primary percutaneous coronary intervention (pPCI). In the early infarction period, the management and treatment of patients with STEMI MINOCA and MI-CAD differ substantially. Primary PCI is a key component of treatment in STEMI MI-CAD and is closely related to short- and long-term prognosis, while this is of less relevance in STEMI MINOCA patients. Therefore, coronary angiography is only an exclusive tool for significant atherosclerosis and could be helpful in diagnosing other conditions alike spontaneous coronary dissection. Clinical characteristics of MINOCA patients differ significantly from all other patients with AMI. These individuals are more frequently younger, while the disease affects both sexes equally, unlike MI-CAD. Patients with MINOCA tend to have fewer cardiovascular risk-factors, and their serum markers of myocardial damage peak at lower levels [1–9].

The aim of our study was to develop a score that allows to distinguish between STEMI patients with MINOCA and MI-CAD at admission to hospital and to apply appropriate therapy in the early MI period. The secondary objective was to describe the characteristics of STEMI patients with MINOCA and MI-CAD.

## Materials and methods

### Study design and patient population

This retrospective analysis was performed on prospectively collected data. Data for conducting the current study were obtained from the Polish National Registry of Percutaneous Coronary

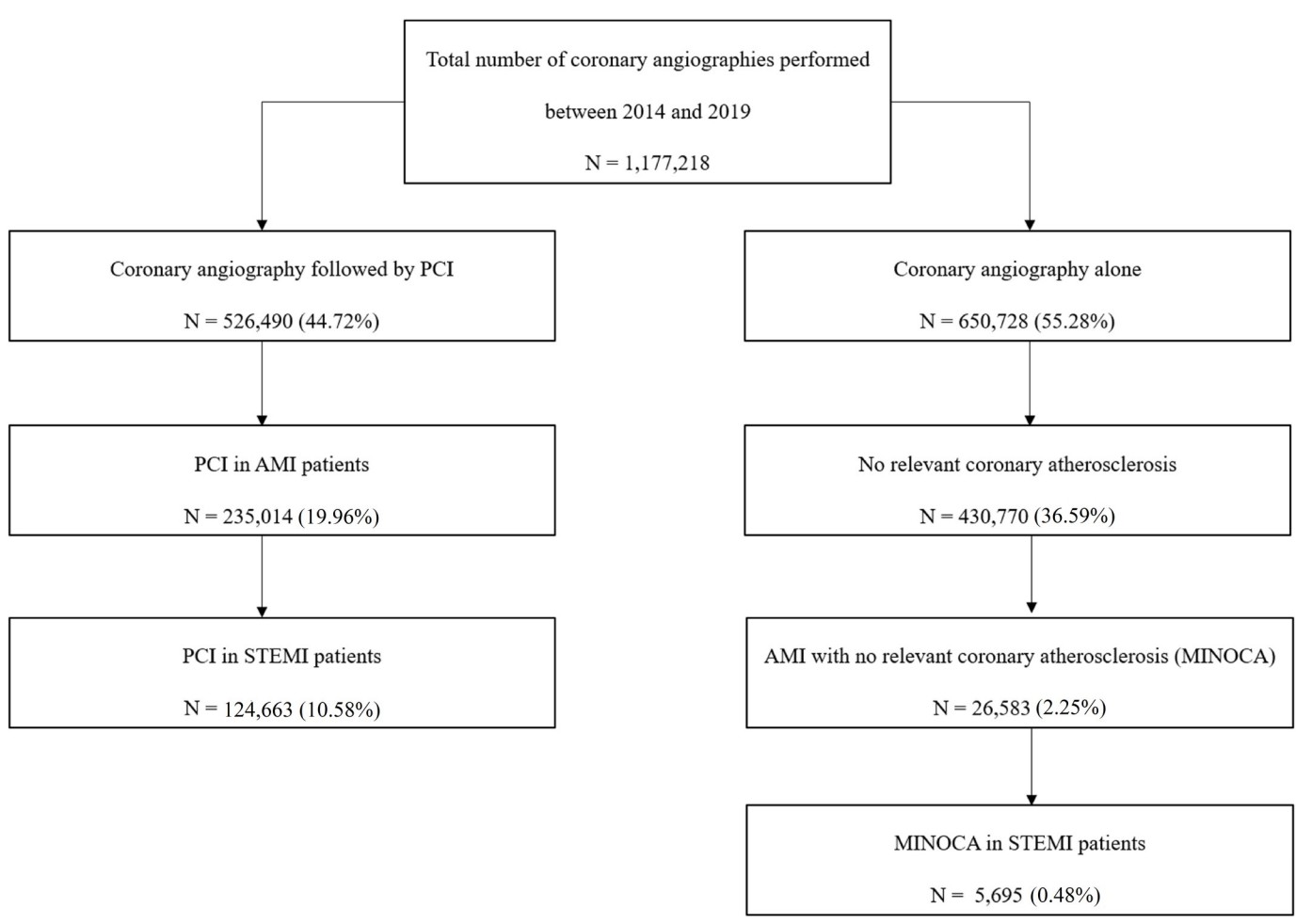

**Fig 1. Patient flow chart.**

Interventions (ORPKI) [10–12]. Data were collected between January 2014 and December 2019, and were selected from 1,177,218 patients who underwent coronary angiography. We classified patients into 2 groups: STEMI MINOCA and STEMI MI-CAD. We selected 124,663 subjects treated due to STEMI out of 526,490 treated using PCI during the analysed period, and 5,695 STEMI patients with MINOCA out of 430,770 with no relevant coronary atherosclerosis in coronary angiography (Fig 1). The diagnosis of STEMI was established by operators working in the catheterisation laboratory according to applicable European Guidelines [13–15]. While suspected diagnosis of MINOCA was established after exclusion of atherosclerotic lesions or significant atherosclerosis (>50% diameter stenosis in the major epicardial artery) which was, in selected cases, supported by other more sophisticated imaging studies, such as intravascular ultrasound and optical coherence tomography [16]. Further elimination of patients from the suspected MINOCA group who, ultimately, did not have a diagnosis of MINOCA (e.g. myocarditis or stress cardiomyopathy), was carried out after discharge from the catheterisation laboratory. Therefore, in the current study, we analysed a heterogenous cohort of 'suspected MINOCA'. Technical aspects of the procedure, i.e. the choice of access site (femoral or radial sheath), catheter size, guidewires, type of thrombectomies and other devices, were at the operator's discretion and complied with the relevant European Guidelines [13–15]. Furthermore, periprocedural anticoagulation and indications for PCI, as well as stent

type, also remained at the first operator's discretion. Antiplatelet therapy was implemented according to current European Guidelines [17]. The diagnosis of no significant stenosis was based on coronary angiography alone (stenosis less than 50% diameter) or in selected and borderline lesions was followed by more sophisticated imaging (intravascular ultrasound or optical coherence tomography) or physiological (fractional flow reserve) tests, conducted according to current European Guidelines [18]. The protocol complied with the 1964 Declaration of Helsinki and all participants provided their written informed consent to participate in the percutaneous intervention. Due to the retrospective nature of the study, as well as anonymisation of the collected data and registry, obtaining consent of the Bioethics Committee was not required.

## Endpoints

Primary study endpoints of the presented analysis included predictors of MINOCA patients among the overall group of subjects admitted to hospital with a STEMI diagnosis and qualified for diagnostic coronary angiography. Additionally, we aimed to create a MINOCA score that could help in separating patients with MINOCA from those with MI-CAD among the overall group of STEMI patients admitted to hospital for diagnostic coronary angiography. Furthermore, our objective was to indicate differences in clinical features, medical treatment, coronary angiography as well as the duration until treatments between MINOCA and MI-CAD patients admitted to hospital with STEMI and qualified for coronary angiography diagnostics.

## Statistical analysis

Categorical variables were expressed as counts and percentages. The chi-squared test or Fisher's exact test were used to identify differences between the MINOCA and MI-CAD groups. Continuous variables are expressed as means (SD) or medians (Q1-Q3). The Student's $t$-test or the Wilcoxon sum rank test were used to identify significant differences, where applicable. The initially analysed predictors included all variables showing a $P$ value of less than .2. To develop the final model, the backward step-down selection method was performed with minimisation of the Bayesian Information Criterion (BIC) as a target. Classification was presented using the receiver operating characteristic (ROC) with the area under the curve (AUC). The final model was built on the entire sample of patients, hence, to evaluate the model performance, internal model validation was performed by comparison of the C-statistic, calculated from the model to the C-statistic using bootstrapping with 1,000 replications. Each of the 1,000 bootstrap samples contained a randomly selected (with replacement) set of patients and a separate model was created with the same set of covariates for each sample. Predictive performances of all models were pooled into 1 bias-corrected C-index. Covariate distributions were superimposed on nomogram scales, which resulted in regression coefficients being represented on a 0 to 100 scale. These scores were then totalled and mapped onto the probability scale to the associated total number of points with the probability of MINOCA. The nomogram was created on the basis of the final, validated model. All statistical tests were two-sided, with a significance level of .05. Data management and statistical analyses were performed using JMP 15.2.0 (SAS Institute Inc, Cary, NC, USA, 2020) and R 4.0.2 (R Foundation for Statistical Computing, Vienna, Austria, 2020).

## Designing the score for suspected MINOCA

In order to design the MINOCA predictive score, using multivariate analysis, we first calculated predictors of MINOCA vs. MI-CAD in STEMI patients. The score for suspected MINOCA was created applying the previous regression model, in which the calculated coefficients

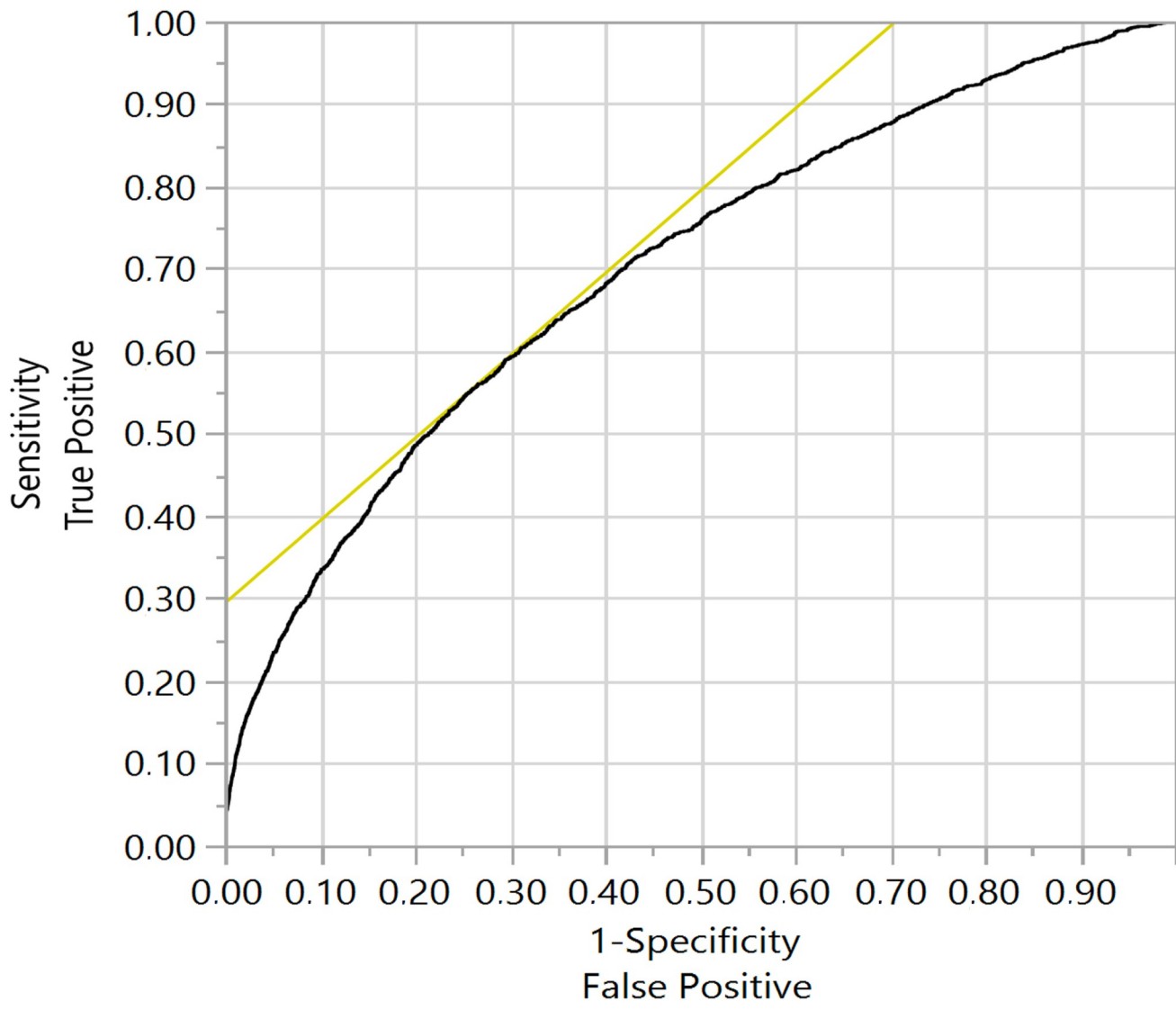

**Fig 2. Receiver operating characteristics.** Using STEMI MINOCA—1 vs. STEMI MI-CAD + PCI– 0 = '1' to be the positive level. Area under the curve—0.7.

in the final model were implemented to construct a predictive model in the form of a nomogram, that is, a graphical representation regarding relative impact of each prognostic factor within the global model (Figs 2 and 3). The model allowed to demonstrate that patients who scored more than 600 points had a 19% probability of MINOCA, while for patients scoring more than 650 points, the probability was 71%. On the other end of the scale, MINOCA probability was marginal for patients who scored less than 500 points (< .2%).

## Results

### Clinical characteristics at baseline

Patients with MINOCA differ significantly from those exhibiting MI-CAD. They were significantly younger, less often males, and did not suffer from diabetes, arterial hypertension or kidney failure as frequently. They were also less often smokers, however, the occurrence of

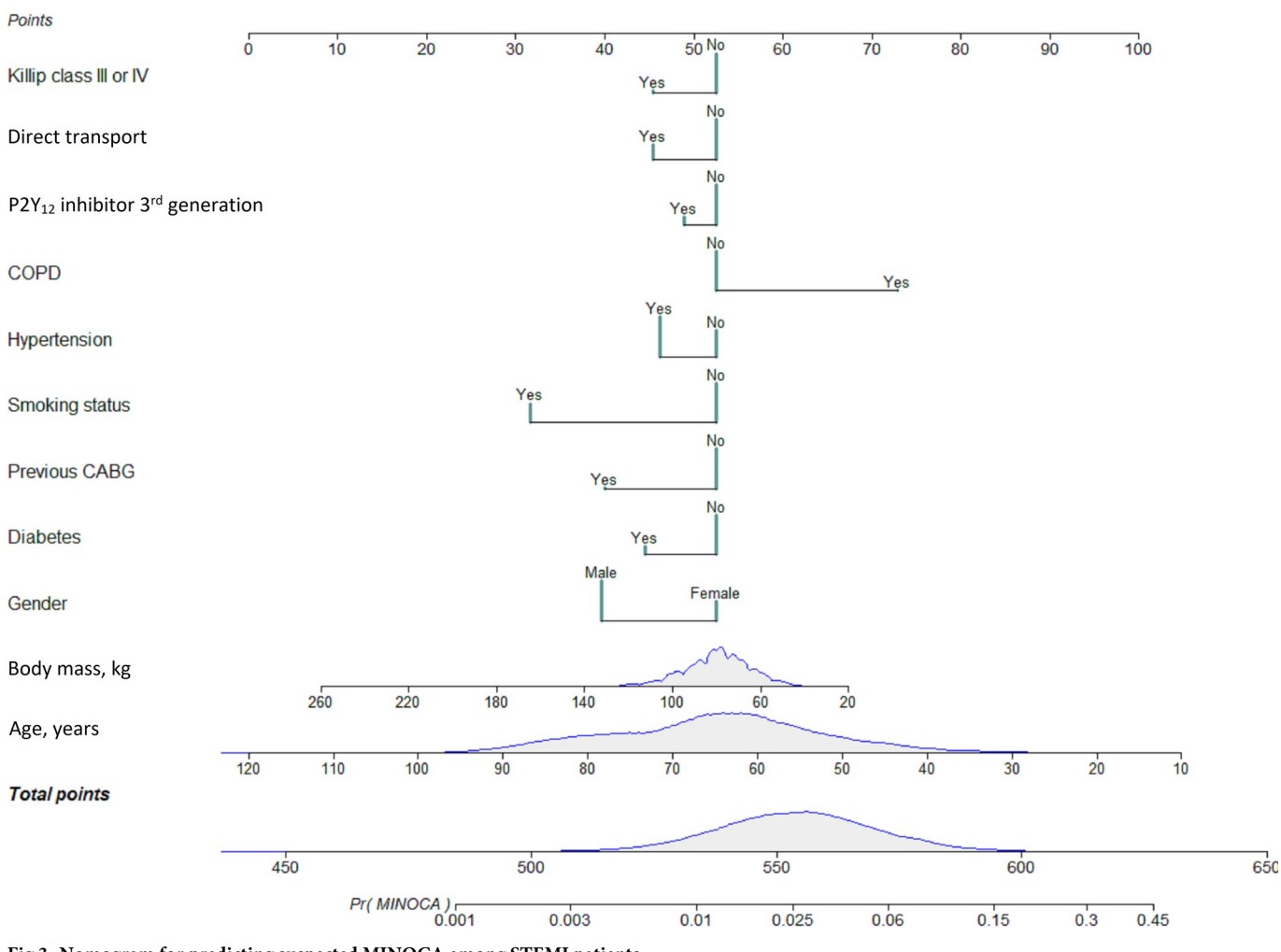

**Fig 3. Nomogram for predicting suspected MINOCA among STEMI patients.**

chronic obstructive pulmonary disease was greater in comparison to the MI-CAD patients. Subjects from the MINOCA group were also less often found to have suffered prior myocardial infarction, PCI, coronary artery bypass grafting or stroke (Table 1).

## Coronary angiography and procedural indices

Considering patients from the MINOCA group, 71.57% had no significant stenoses. while 28.43% exhibited no visible signs of atherosclerosis. In the MI-CAD group, there was a comparable frequency of patients with single- (45.71%) and multi-vessel disease, without left main coronary artery involvement (47.35%). Patients in the MINOCA group were significantly more frequently treated from radial than femoral access compared to the MI-CAD group ($P < .01$). Intravascular ultrasound was significantly more often applied in the MI-CAD, while fractional flow reserve and optical coherence tomography were used in the MINOCA group (Table 2).

## Pharmacotherapy, transport data and procedure-related complications

Patients from the MINOCA group were significantly less often treated with antiplatelets and heparins when compared to MI-CAD subjects (Table 3). Hypothermia was also more

**Table 1. Clinical characteristics at baseline in STEMI patients treated with PCI according to presence of significant coronary atherosclerosis (MI-CAD vs. MINOCA).**

| Variables | Total N = 130,358 | MI-CAD N = 124,663 | MINOCA N = 5,695 | P |
|---|---|---|---|---|
| Age, years | 65.0 ± 12.3 | 65.2 ± 12.1 | 60.7 ± 15.9 | < .001 |
| Body mass, kg | 80.1 ± 16.5 | 80.3 ± 16.5 | 77.1 ±18.1 | < .001 |
| Gender, male | 87,210 (67.3) | 84,003 (67.8) | 3,207 (57.1) | < .001 |
| Diabetes | 22,801 (17.5) | 22,213 (17.8) | 588 (10.3) | < .001 |
| Arterial hypertension | 76,135 (58.4) | 73,556 (59) | 2,579 (45.3) | < .001 |
| Prior stroke | 4,188 (3.2) | 4.051 (3.2) | 137 (2.1) | < .001 |
| Prior myocardial infarction | 16,085 (12.3) | 15,553 (12.5) | 532 (9.3) | < .001 |
| Prior PCI | 14,958 (11.5) | 14,431 (11.6) | 527 (9.2) | < .001 |
| Prior CABG | 2,255 (1.7) | 2,188 (1.8) | 67 (1.2) | < .001 |
| Smoking | 38,311 (29.4) | 37,326 (29.9) | 985 (17.3) | < .001 |
| Kidney failure | 4,381 (3.4) | 4,231 (3.4) | 150 (2.6) | .002 |
| Chronic obstructive pulmonary disease | 2,201 (1.7) | 2,076 (1.7) | 125 (2.2) | .003 |
| Killip class | | | | < .001 |
| Mean ± SD | 1.3 ± 0.7 | 1.3 ± 0.7 | 1.2 ± 0.6 | |
| • I | 91,284 (82.9) | 87,266 (82.6) | 4,018 (88.5) | |
| • II | 11,118 (10.1) | 10,826 (10.2) | 292 (6.4) | |
| • III | 3,450 (3.1) | 3,342 (3.2) | 108 (2.4) | |
| • IV | 4,314 (3.9) | 4,191 (4.0) | 123 (2.7) | |

Data are presented as means ± standard deviation or counts (percentages); percentages reflect available study data.

CABG: Coronary artery bypass grafting; PCI: Percutaneous coronary intervention.

frequently noted in the MI-CAD group. Considering time lengths, direct transport to the catheterisation laboratory was noted significantly more often in MI-CAD patients when compared to MINOCA, as well as other delays in treatment rates (time from pain to first contact, to inflation or angiogram, and from first contact to inflation or angiogram) were longer in the MINOCA group compared to MI-CAD (Table 3). Periprocedural death and cardiac arrest rates were significantly higher in the MI-CAD patients in comparison to the MINOCA group (Table 3).

## Predictors of MINOCA vs. MI-CAD patients admitted due to STEMI– univariate analysis

All significant predictors of MINOCA vs. MI-CAD in STEMI patients assessed by univariate analysis are presented in Table 4, while those evaluated via multivariate analysis are included in Table 5.

## Discussion

In the current study, significant differences were revealed in clinical image, procedural features, pharmacotherapy and duration until treatments between MINOCA and MI-CAD patients among those from the STEMI group qualified for urgent coronary angiography. Among others, patients with MINOCA were significantly younger, with a smaller burden of concomitant diseases and better clinical state assessed by Killip class grade at admission. They were significantly more frequently treated from radial access, while less often treated with anti-platelets and unfractionated heparins. We confirmed younger age, female gender, no history of prior CABG or smoking, no arterial hypertension, presence of COPD, no treatment with

**Table 2. Coronary angiography and culprit lesion characteristics in STEMI patients treated with PCI according to presence of significant coronary atherosclerosis (MI-CAD vs. MINOCA).**

| Variables | Total N = 130,358 | MI-CAD N = 124,663 | MINOCA N = 5,695 | P |
|---|---|---|---|---|
| Coronary angiography: | | | | < .001 |
| No significant stenoses | 4,076 (3.1) | 0 (0) | 4,076 (71.6) | |
| No visible atherosclerosis | 1,619 (1.2) | 0 (0) | 1,619 (28.4) | |
| Single-vessel disease | 56,905 (43.7) | 56,905 (45.7) | 0 (0) | |
| MVD without LMCA | 58,943 (45.3) | 58,943 (47.3) | 0 (0) | |
| MVD and LMCA | 8,344 (6.4) | 8,344 (6.7) | 0 (0) | |
| Isolated LMCA | 300 (0.23) | 300 (0.24) | 0 (0) | |
| Vascular access | | | | < .001 |
| Femoral | 34,801 (26.71) | 33,548 (26,92) | 1,253 (22.04) | |
| Radial left | 18,334 (14.07) | 17,586 (14.11) | 748 (13.16) | |
| Radial right | 76,450 (58.7) | 72,793 (58.4) | 3,657 (64.3) | |
| Others | 708 (0.5) | 680 (0.5) | 28 (0.5) | |
| Radiation dose, Gy | 998 ± 954.6 | 1028.3 ± 961.8 | 319.7 ± 347 | < .001 |
| | 737 [409 ÷ 1270] | 766 [437 ÷ 1301] | 235 [133 ÷ 400] | |
| Contrast amount, mL | 171.1 ± 75.5 | 175.4 ± 74.1 | 79.1 ± 39.8 | < .001 |
| | 150 [120 ÷ 200] | 150 [120 ÷ 200] | 70 [50 ÷ 100] | |
| Additional imaging tests during PCI | | | | |
| Fractional flow reserve | 139 (0.1) | 89 (0.07) | 50 (0.88) | < .001 |
| Intravascular ultrasound during | 288 (0.2) | 255 (0.2) | 33 (0.6) | < .001 |
| Optical coherence tomography | 66 (0.05) | 57 (0.05) | 9 (0.2) | < .001 |

Data are presented as mean ± standard deviation, medians [lower ÷ upper quartile] for asymmetric distribution or counts (percentages); percentages reflect available study data.

LMCA: Left main coronary artery; MVD: Multi-vessel disease; PCI: Percutaneous coronary intervention.

$3^{rd}$ generation P2Y$_{12}$, no direct transport to hospital, Killip class I or II, no history of diabetes and greater body mass at admission to be significant predictors of MINOCA among STEMI patients qualified for urgent coronary angiography. Using the MINOCA score model, we were able to find that in patients scoring above 600 points, the probability of MINOCA was 19%, with a score above 650 points. This probability was more than 3-fold (71%), while for patients scoring less than 500 points, the probability was considered marginal (< .2%).

The main objective of creating a tool facilitating rapid separation of MINOCA patients from the STEMI entire group urgently admitted to hospital is to distinguish a group of patients who do not require urgent invasive interventions, associated with the possibility of various complications. In order to create such a tool, researchers have attempted to enumerate the features of MINOCA patients that are typical for this group. Clinical features of MINOCA patients depend on the analysed group of patients, including type of MI, and they differ between published studies. However, there are some common features independent from analysis. The data obtained by Ballesteros-Ortega were based on an overall group of AMI, including 9,241 MI-CAD and 622 MINOCA patients [19]. The main characteristics of the MINOCA patients were as follows: younger age than patients with obstructive coronary disease, but not differing in distribution according to gender, fewer risk-factors for cardiovascular disease or known cardiovascular conditions before admission, lower disease severity during admission and, therefore, a lower rate of complications, including death. The authors found 8 factors among those classified as significant predictors of MINOCA and included them in the normogram for predicting MINOCA: prior AMI, peripheral arterial disease, diabetes mellitus,

**Table 3. Pharmacotherapy, periprocedural complications and transport data in STEMI patients treated with PCI according to presence of significant coronary atherosclerosis (MI-CAD vs. MINOCA).**

| Variables | Total N = 130,358 | MI-CAD N = 124,663 | MINOCA N = 5,695 | P |
|---|---|---|---|---|
| Pharmacotherapy | | | | |
| Acetyl-salicylic acid during angiogram | 92,346 (70.8) | 88,954 (71.4) | 3,392 (59.6) | < .001 |
| Unfractionated heparin during angiogram | 65,477 (50.2) | 62,982 (50.5) | 2,495 (43.8) | < .001 |
| Low molecular weight heparin during angiogram | 72 (0.06) | 7 (0.01) | 65 (1.14) | < .001 |
| P2Y$_{12}$ during angiogram | | | | < .001 |
| Clopidogrel | 62,975 (48.3) | 60,724 (48.7) | 2,251 (39.5) | |
| Ticagrelor | 11,396 (8.7) | 11,061 (8.9) | 334 (5.9) | |
| Prasugrel | 549 (0.4) | 535 (0.4) | 14 (0.2) | |
| No-dual antiplatelet therapy | 55,438 (42.5) | 52,342 (42) | 3,096 (54.4) | |
| Glycoprotein IIb/IIIa inhibitor during angiogram | 30,651 (23.5) | 30,646 (24.6) | 5 (0.1) | < .001 |
| Hypothermia at baseline | 253 (0.2) | 247 (0.2) | 6 (0.11) | .08 |
| Transport data | | | | |
| Direct transport | 31,202 (23.9) | 30,263 (24.3) | 939 (16.5) | < .001 |
| Time from pain to first contact, min. | 564.5 ± 4566.3 | 563.93 ± 4635.5 | 579.3 ± 2262.4 | < .001 |
| | 120 [60 ÷ 330] | 120 [60 ÷ 330] | 134 [60 ÷ 408] | |
| Time from pain to inflation or angiogram, min. | 1009.3 ± 9094.4 | 998.2 ± 8983.3 | 1389.1 ± 12305.5 | < .001 |
| | 240 [145 ÷ 532] | 240 [145 ÷ 529] | 300 [150 ÷ 655.2] | |
| Time from first contact to inflation or angiogram, min. | 468.5 ± 7916.7 | 458.67 ± 7791.9 | 808.7 ± 11431.2 | < .001 |
| | 87 [60 ÷ 145] | 86 [60 ÷ 143] | 99 [60 ÷ 191] | |
| Procedure-related complications | | | | |
| Puncture-site bleeding | 59 (0.05) | 57 (0.05) | 2 (0.04) | .71 |
| Cardiac arrest | 1,000 (0.8) | 986 (0.8) | 14 (0.2) | < .001 |
| Allergic reaction | 14 (0.01) | 13 (0.01) | 1 (0.02) | .64 |
| Death | 1,611 (1.2) | 1,597 (1.3) | 14 (0.2) | < .001 |

Data are presented as mean ± standard deviation, medians [lower ÷ upper quartile] for asymmetric distribution or counts (percentages); percentages reflect available study data.

hypertension, smoking, troponin elevation (9 times above the upper norm), age above 50 and gender [19]. In their study, Roe et al. attempted to differentiate between ACS patients without STEMI but having obstructive disease, and those who did not [20]. The authors designed a score based on the patients' baseline characteristics, which could be successfully used to predict the presence of coronary artery disease, without significant obstruction. Following, the authors applied their predictive model to patients in the GUSTO-IIb trial (Global Use of Strategies to Open Occluded Coronary Arteries in Acute Coronary Syndromes), demonstrating its surprisingly high usefulness in differentiating patients with the diagnoses under study [21].

In our research, we created a score to predicting suspected MINOCA vs. MI-CAD among patients hospitalised due to MI with ST segment elevation. The score for predicting suspected MINOCA was created to assess patients with myocardial infarction, having non-obstructive coronary arteries. There is no such tool that could be used to predict the risk of MINOCA occurrence among patients admitted to hospital with MI-ST-elevation segment, which would be useful in preparing an investigation plan necessary to design an individualised treatment course for the MINOCA group. Further investigation is obligatory among MINOCA patients according to the results presented in the ESC MINOCA position paper [14]. They allow to indicate that patients with MI-CAD were more often directly transported to a cardiac catheterisation laboratory than those treated via MINOCA. This may mainly be explained by higher

**Table 4. Predictors of MINOCA vs. MI-CAD in patients admitted due to STEMI–univariate analysis.**

| Selected factor | Odds ratio | 95% confidence interval | P |
|---|---|---|---|
| Gender, female | 1.58 | 1.49–1.66 | < .001 |
| Diabetes | 0.53 | 0.48–0.57 | < .001 |
| Prior stroke | 0.73 | 0.61–0.87 | < .001 |
| Prior myocardial infarction | 0.72 | 0.66–0.79 | < .001 |
| Prior coronary artery bypass grafting | 0.66 | 0.52–0.85 | .001 |
| Smoking | 0.48 | 0.45–0.52 | < .001 |
| Arterial hypertension | 0.57 | 0.54–0.6 | < .001 |
| Kidney failure | 0.77 | 0.65–0.9 | .001 |
| Chronic obstructive pulmonary disease | 1.32 | 1.1–1.59 | .003 |
| Acetyl-salicylic acid | 0.59 | 0.56–0.62 | < .001 |
| Unfractionated heparin | 0.76 | 0.72–0.8 | < .001 |
| Low-molecular weight heparin | 359.8 | 131.04–987.92 | < .001 |
| $P2Y_{12}$ inhibitor | | | |
| Prasugrel vs. ticagrelor | 0.86 | 0.5–1.48 | < .001 |
| Ticagrelor vs. clopidogrel | 0.81 | 0.72–0.91 | < .001 |
| Prasugrel vs. clopidogrel | 0.7 | 0.41–1.2 | < .001 |
| Cardiac arrest at baseline | 0.84 | 0.74–0.96 | .009 |
| Direct transport | 0.61 | 0.57–0.66 | < .001 |
| Killip class II vs. I | 0.58 | 0.51–0.66 | < .001 |
| Killip class III vs. I | 0.7 | 0.57–0.85 | < .001 |
| Killip class IV vs. I | 0.63 | 0.53–0.76 | < .001 |

values of Killip class and more severe changes in ECG, which maintain in agreement with the results obtained by Ballesteros-Ortega [19]. Patients from the MINOCA group statistically less frequently received antiplatelet medication prior to admission, however, they obviously received HNF during coronary angiography.

What is interesting, in studies on the use of DAPT in MINOCA patients, it is shown that therapy is not only of no benefit, but may be detrimental in this group of patients [22]. Additionally, it should be noted that, so far, possible harm associated with DAPT is mainly related to higher major bleeding rates, given also the higher incidence of MINOCA in women [23]. In

**Table 5. Nominal logistic fit for MINOCA vs. MI-CAD patients admitted due to STEMI–multiple regression analysis.**

| Factor | Odds ratio | 95% confidence interval | P |
|---|---|---|---|
| Age, years | 0.96 | 0.95–0.96 | < .001 |
| Gender, female | 1.83 | 1.65–2.02 | < .001 |
| Prior CABG | 0.57 | 0.34–0.95 | .03 |
| Smoking | 0.38 | 0.33–0.42 | < .001 |
| Arterial hypertension | 0.74 | 0.68–0.82 | < .001 |
| COPD | 2.5 | 1.89–3.3 | < .001 |
| $P2Y_{12}$ 3rd generation | 0.84 | 0.74–0.95 | .008 |
| Direct transport | 0.72 | 0.64–0.8 | < .001 |
| Killip class III or IV | 0.72 | 0.57–0.89 | .003 |
| Diabetes | 0.69 | 0.59–0.79 | < .001 |
| Body mass, kg | 0.98 | 0.98–0.99 | < .001 |

Test and confidence intervals on odds ratios are Wald-based.

this regard the post-hoc analysis of the OASIS-7 trial very nicely inform us about the possible harm related to intensified dosing strategy [23].

The results of the analysed data presented in the current publication, refer to the pre-hospital period, often that directly pre-hospital, therefore, it is not possible to assess the impact on prognosis of antiplatelet drug amount during the follow-up period. Certainly, in some patients, DAPT therapy is associated with recent PCI or acute coronary syndrome in recent months, but it is not possible to estimate its percentage on the basis of the data available in the registry.

According to the results of our study, co-morbidities, especially hypertension, diabetes or previously existing ischemic heart disease with post-coronary artery bypass graft, are negative predictors of MINOCA. Analogous results can be found in the VIRGO study and a large systematic review of MINOCA patients by Pasupathy et al. [2,24]. Among the other negative predictive factors of MINOCA, we may distinguish being overweight. Patients from the MI-CAD group were more often obese compared to MINOCA. This was also confirmed in other research trials on MINOCA. As it is well-known, obesity remains one of the most significant risk-factors for developing atherosclerosis. MINOCA patients are more frequently non-smokers, maybe because of the greater number of women in the this group. Despite the increasing number of women smoking tobacco, men are still more likely to be smokers. Interestingly, COPD turned out to be a positive predictor of MINOCA. This was not confirmed in any other study and definitely requires further investigation.

The main limitation of our study is the lack of other significant parameters, such as those related to echocardiography, high sensitive troponin levels, C-reactive protein levels, red and white cell concentrations or BNP, which may have significant impact on our scoring system. These parameters may definitely increase the predictive value of our score.

## Strengths and limitations

Considering the major limitations of the current analysis, the leading one may the registry nature, and thus, bias—mainly caused by lack of core laboratory and subjective data collection by the operators. Another key element is the lack of a large pool of data, including laboratory, electrocardiography, echocardiography and other imaging tests which could certainly change the composition of individual factors included in the MINOCA score. The data provided in the present publication come from a registry in which only periprocedural data is collected, and is further supplemented by operators of catheterisation ambulatories. Therefore, we do not possess data on the final etiology of MINOCA. It is already known that the establishment of MINOCA etiology, in a great percentage of cases, is possible, even many weeks after the initial incident, which is conditioned by the need to perform additional tests, such as a thrombophilia test package, cardiac magnetic resonance imaging or other additional diagnostics tests. Therefore, in the analysed group of patients, there are also subjects in whom the diagnosis of MINOCA has not been ultimately confirmed. However, when it comes to finding coronary spasm or dissection, including spontaneous coronary dissection, we have access to such data, but because of the large percentage of missing data, we have refrained from including it in statistical calculations due to the high probability of bias and drawing erroneous conclusions.

On the other hand, the unquestionable advantage of the studied registry is the large number of patients and its real-life nature, rather than being a programmed trial.

## Conclusions

In the current study, significant differences were revealed regarding clinical image, procedural features, pharmacotherapy and duration until treatments between MINOCA and MI-CAD patients among STEMI patients qualified for urgent coronary angiography. Based on the

MINOCA score created in the current publication, we may be tempted to distinguish, with moderate probability, MINOCA from MI-CAD patients in the STEMI group qualified for urgent coronary angiography. This possibility is based on the predictors calculated via multivariate analysis of the available indices in the current study. Specifying the MINOCA score by including more data on an equal or greater number of subjects in the calculation, which is not available in the database analysed in this publication, may contribute to the creation of such a MINOCA score, thanks to which it would be very likely to avoid unnecessary urgent coronary angiography among MINOCA patients in those with STEMI.

In conclusion, the need to decrease the number of unjustified angiograms in STEMI patients is undeniable. However, finding clinical approval for doing so, especially on the basis of the existing results, is not a simple task. Nonetheless, potentially applying such predictive algorithms may aid operators in preparation for MINOCA diagnosis, and as a result, further testing. Moreover, in to date, there are not supplementary diagnostic tools that could be helpful or useful in MINORCA management. It is often the case that patients are not able to comprehend their presentation, and are consequently discharged from hospital without a concrete diagnosis. Debate and examination of possible MINORC diagnosis could significantly help these patients obtain a better understanding of their condition. For these reasons, the application of predictive algorithms in MINOCA patients does have its position. Nonetheless, the derived population should be defined accordingly, while tool implementation should be concentrated on improving communication and management in the system of health care.

## Author Contributions

**Conceptualization:** Magdalena Jędrychowska, Zbigniew Siudak, Piotr Mika, Krzysztof Bartuś, Wojciech Wańha, Wojciech Wojakowski, Stanisław Bartuś, Rafał Januszek.

**Data curation:** Zbigniew Siudak.

**Formal analysis:** Krzysztof Piotr Malinowski, Rafał Januszek.

**Investigation:** Magdalena Jędrychowska, Łukasz Zandecki, Michał Zabojszcz, Tomasz Kameczura, Piotr Mika, Wojciech Wojakowski, Rafał Januszek.

**Methodology:** Magdalena Jędrychowska, Zbigniew Siudak, Tomasz Kameczura, Wojciech Wańha, Rafał Januszek.

**Project administration:** Zbigniew Siudak, Krzysztof Bartuś, Rafał Januszek.

**Software:** Krzysztof Piotr Malinowski.

**Supervision:** Zbigniew Siudak, Krzysztof Bartuś, Wojciech Wańha, Wojciech Wojakowski, Jacek Legutko, Stanisław Bartuś, Rafał Januszek.

**Validation:** Magdalena Jędrychowska, Zbigniew Siudak, Jacek Legutko, Rafał Januszek.

**Visualization:** Magdalena Jędrychowska, Rafał Januszek.

**Writing – original draft:** Magdalena Jędrychowska, Łukasz Zandecki, Michał Zabojszcz, Tomasz Kameczura, Piotr Mika, Rafał Januszek.

**Writing – review & editing:** Zbigniew Siudak, Krzysztof Bartuś, Wojciech Wańha, Wojciech Wojakowski, Jacek Legutko, Stanisław Bartuś, Rafał Januszek.

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
