## [Decision Letter · Decision Letter 0]

12 Jan 2021

PONE-D-20-39776

ST-segment elevation myocardial infarction with non-obstructive coronary arteries: derivation of a score for prediction based on a large national registry

PLOS ONE

Dear Dr. Januszek,

Thank you for submitting your manuscript to PLOS ONE. After careful consideration, we feel that it has merit but does not fully meet PLOS ONE’s publication criteria as it currently stands. Therefore, we invite you to submit a revised version of the manuscript that addresses the points raised during the review process.

We look forward to receiving your revised manuscript.

Kind regards,

Carmine Pizzi

Academic Editor

PLOS ONE

2. You indicated that ethical approval was not necessary for your study. We understand that the framework for ethical oversight requirements for studies of this type may differ depending on the setting and we would appreciate some further clarification regarding your research. Could you please provide further details on why your study is exempt from the need for approval and confirmation from your institutional review board or research ethics committee (e.g., in the form of a letter or email correspondence) that ethics review was not necessary for this study? Please include a copy of the correspondence as an ""Other"" file.

3.In your Data Availability statement, you have not specified where the minimal data set underlying the results described in your manuscript can be found. PLOS defines a study's minimal data set as the underlying data used to reach the conclusions drawn in the manuscript and any additional data required to replicate the reported study findings in their entirety. All PLOS journals require that the minimal data set be made fully available. For more information about our data policy, please see http://journals.plos.org/plosone/s/data-availability.

4.Thank you for stating the following financial disclosure:

 "NO - The funders had no role in study design, data collection and analysis, decision to publish, or preparation of the manuscript."

5.We noticed you have some minor occurrence of overlapping text with the following previous publication(s), which needs to be addressed:

https://www.dovepress.com/characteristics-of-patients-with-myocardial-infarction-with-nonobstruc-peer-reviewed-article-VHRM

https://www.mdpi.com/2077-0383/9/11/3610/html

The text that needs to be addressed involves sections of the Introduction and Discussion.

In your revision ensure you cite all your sources (including your own works), and quote or rephrase any duplicated text outside the methods section. Further consideration is dependent on these concerns being addressed."

Reviewers' comments:

Reviewer #1: In this paper Jędrychowska et al. create a MINOCA STEMI predictive score using the regression model, in order to distinguish, in STEMI group, MINOCA patients from MI-CAD patients; younger age, female gender, no history of prior CABG, no history of smoking, no arterial hypertension, presence of COPD, no treatment with 3rd generation P2Y12, no direct transport to hospital, Killip class I or II, no history of diabetes and greater body mass at admission were considered significant predictors of MINOCA among STEMI patients qualified for urgent coronary angiography and used to develop the score. The coefficients calculated were used to construct the predictive model in the form of a nomogram. The model showed that patients who scored more than 600 points had a 19% probability of MINOCA, and for patients scoring more than 650 points, the probability of MINOCA was 71%.

The study is interesting and focuses on a relatively new diagnostic problem such as the MINOCA patients. Nevertheless, I have some major concerns:

1) MINOCA diagnosis  according to ESC Guidelines?  how do you exclude the cases of non-ischemic troponin elevation? These patients are not MINOCA (3° point of ESC Position Paper).

2) There are no data on troponin. How was STEMI diagnosed?

3) Table 1: The majority of MINOCA patients with prior AMI, had also prior PCI  so they supposed to have a significative CAD. How these patients could be classified as MINOCA?

4) Table 1: 67 patients classified as MINOCA had a previous CABG. The same as before, these patients had significative CAD. I suppose they are not MINOCA.

5) Table 2: “Considering patients from the MINOCA group, 71.57% had significant stenoses while 28.43% no visible atherosclerosis”. If these patients had significant CAD, why they are called MINOCA?

6) What was the cause of MINOCA? Is there any information for the prevalence of plaque disruption, coronary artery spasm, coronary thromboembolism, coronary dissection?

7) It is sufficient to approximate the numbers in the tables to the first decimal. Tables will be more readable.

8) 45% of MINOCA are in DAPT. In these patients, was DAPT therapy administered even without DES implantation?

9) The score assigned to each variable is not clear.

10) The main purpose is understanding how distinguish a priori a patient with MINOCA and possibly not to perform a coronary study. I believe that the results are absolutely not suitable for reaching this conclusion. Even if the score is > 650, the probability that it is a MINOCA is about 71%; therefore, it means that about 30% of patients are not MINOCA, actually; I don't think with these numbers it is possible to decide not to perform a CAG to a patient when it is recognized that an early PTCA can improve the prognosis. The criteria for performing a CAG as soon as possible (in STEMI) have been formulated precisely because they admit a low sensitivity and therefore provide for the possibility of having false positives with patients with free coronary arteries.

11) AUC 0.7  It seems to me little bit low.

Reviewer #2: M. Jędrychowska and coworkers studied the main clinical and angiography differences in a large population of STEMI in both obstructive acute myocardial infarction and MINOCA patients. Data were collected between January 2014 and December 2019, and were selected from 1,177,218 patients who underwent coronary angiography. The authors aimed to create a tool facilitating a rapid separation of the MINOCA patients from the entire group of obstructive STEMI ones.

Finally, they evaluated a score able to address and separate MINOCA from obstructive AMI patients hospitalized due to MI with ST segment elevation. The model showed that patients who scored more than 600 points had a 19% probability of MINOCA, and for patients scoring more than 650 points, the probability of MINOCA was 71%. The other end of the MINOCA probability scale was marginal for patients who scored less than 500 points (< .2%).

The strengths of this study are:

- First, the large population of the study. Indeed they evaluated more than 1 million patients from the Polish National Registry of Percutaneous Coronary Interventions (ORPKI). They selected a broad sample of STEMI underwent PCI and MINOCA with ST-segment elevation (more than 5600 ones). Respect to the current literature on MINOCA, this is a very large population though from a retrospective registry.

- They collected only ST-segment acute myocardial infarction. This is a specific group of patients, especially in the MINOCA, which is worth studying due to higher mortality, particularly intra-hospital.

- The idea of a model able to distinguish STEMI obstructive AMI vs STEMI MINOCA could be very useful in clinical practice due to the subsequent diagnostic and therapeutic implications.

- I appreciate the table 2 in which they showed the coronary angiography and procedural indices.

However, there are some limitations of this study. The major pitfalls are:

- MINOCA patients are a heterogeneous entity. The current literature considered as MINOCA only coronary ischemic causes of acute myocardial infarction without obstructive coronary arteries. In our clinical practice, we have to exclude non-cardiac causes of troponin surge but also cardiac conditions like tako-tsubo syndrome or acute myocarditis.

In this work, the MINOCA population is quite heterogeneous and seemed quite unselected. They had not well-established inclusion and exclusion criteria for MINOCA diagnosis. Moreover, what were the main causes of MINOCA? Did they use cardiac magnetic resonance or other secondary diagnostic tools to address the final MINOCA diagnosis, as the recent guidelines recommend?

- I think that the best clinical utility of the diagnostic model, created in this work, is to rule out the probability of MINOCA diagnosis. In fact, if the score is less than 500 points the probability of MINOCA is quite low < .2%. However, this model could not help the clinicians to avoid an urgent coronary angiography in patients with a suspected STEMI, even if the probability of obstructive CAD is low.

In the paper, the authors should explain more the clinical role of this predictive score (eg: in patients with high MINOCA probability, it’s useful to use more diagnostic tools, like left ventriculography during angiography or intravascular imaging, and to design a specific treatment in this population).

- It could be useful to create a predictive model with only pre-angiography variables, eg exclude P2Y12 inhibitors and add other variables like the presence of typical angina. This could be more useful for the clinicians as explained above.

Minor limitation:

- I suggest a more careful revision of the English language in the text

---

## [Author Response · Author response to Decision Letter 0]

20 Feb 2021

PONE-D-20-39776

ST-segment elevation myocardial infarction with non-obstructive coronary arteries: derivation of a score for prediction based on a large national registry

PLOS ONE

Dear Dr. Januszek,

Thank you for submitting your manuscript to PLOS ONE. After careful consideration, we feel that it has merit but does not fully meet PLOS ONE’s publication criteria as it currently stands. Therefore, we invite you to submit a revised version of the manuscript that addresses the points raised during the review process.

This was modified. 

We look forward to receiving your revised manuscript.

Kind regards,

Carmine Pizzi

Academic Editor

PLOS ONE

 This was modified. 

2. You indicated that ethical approval was not necessary for your study. We understand that the framework for ethical oversight requirements for studies of this type may differ depending on the setting and we would appreciate some further clarification regarding your research. Could you please provide further details on why your study is exempt from the need for approval and confirmation from your institutional review board or research ethics committee (e.g., in the form of a letter or email correspondence) that ethics review was not necessary for this study? Please include a copy of the correspondence as an ""Other"" file.

Due to the fact that the study is conducted on data from the ORPKI register, where the data is entered without names, addresses, telephone numbers or the identification number of individual patients, the consent of the bioethical commission is not required for such research. In addition, one of the works from this database has already been published in the PLOS One Journal, and then, also such documents were not required, which has not changed in this respect.

Januszek R, Dziewierz A, Siudak Z, Rakowski T, Dudek D, Bartuś S. Chronic obstructive pulmonary disease and periprocedural complications in patients undergoing percutaneous coronary interventions. PLoS One. 2018 Oct 1;13(10):e0204257. 

3.In your Data Availability statement, you have not specified where the minimal data set underlying the results described in your manuscript can be found. PLOS defines a study's minimal data set as the underlying data used to reach the conclusions drawn in the manuscript and any additional data required to replicate the reported study findings in their entirety. All PLOS journals require that the minimal data set be made fully available. For more information about our data policy, please see http://journals.plos.org/plosone/s/data-availability.

 The data has been deposited at: Home | OSF/ DOI 10.17605/OSF.IO/MD7AT.

 The data has been deposited at: Home | OSF/ DOI 10.17605/OSF.IO/MD7AT.

Not applicable, simplified version of data has been deposited at : Home | OSF/ DOI 10.17605/OSF.IO/MD7AT.

Not applicable, simplified version of data has been deposited at: Home | OSF/ DOI 10.17605/OSF.IO/MD7AT.

4.Thank you for stating the following financial disclosure:

 "NO - The funders had no role in study design, data collection and analysis, decision to publish, or preparation of the manuscript."

a. Please clarify the sources of funding (financial or material support) for your study. List the grants or organizations that supported your study, including funding received from your institution.

The statement: “The funders had no role in tge study design, data collection, analysis, decision to publish, or preparation of the manuscript” has been added to the Cover Letter.

The statement: “The funders had no role in tge study design, data collection, analysis, decision to publish, or preparation of the manuscript” has been added to the Cover Letter.

 The statement: ”The authors received no specific funding for this work” has been added to the Cover Letter. 

d. If you did not receive any funding for this study, please state: “The authors received no specific funding for this work.”

The statement: ”The authors received no specific funding for this work” has been added to the Cover Letter. 

 This was modified according to the recommendations specified above.

5.We noticed you have some minor occurrence of overlapping text with the following previous publication(s), which needs to be addressed:

https://www.dovepress.com/characteristics-of-patients-with-myocardial-infarction-with-nonobstruc-peer-reviewed-article-VHRM

https://www.mdpi.com/2077-0383/9/11/3610/html

The text that needs to be addressed involves sections of the Introduction and Discussion.

This was rephrased and modified. 

In your revision ensure you cite all your sources (including your own works), and quote or rephrase any duplicated text outside the methods section. Further consideration is dependent on these concerns being addressed."

This was corrected. 

Reviewers' comments:

Reviewer #1: In this paper Jędrychowska et al. create a MINOCA STEMI predictive score using the regression model, in order to distinguish, in STEMI group, MINOCA patients from MI-CAD patients; younger age, female gender, no history of prior CABG, no history of smoking, no arterial hypertension, presence of COPD, no treatment with 3rd generation P2Y12, no direct transport to hospital, Killip class I or II, no history of diabetes and greater body mass at admission were considered significant predictors of MINOCA among STEMI patients qualified for urgent coronary angiography and used to develop the score. The coefficients calculated were used to construct the predictive model in the form of a nomogram. The model showed that patients who scored more than 600 points had a 19% probability of MINOCA, and for patients scoring more than 650 points, the probability of MINOCA was 71%.

The study is interesting and focuses on a relatively new diagnostic problem such as the MINOCA patients. Nevertheless, I have some major concerns:

1) MINOCA diagnosis  according to ESC Guidelines?  how do you exclude the cases of non-ischemic troponin elevation? These patients are not MINOCA (3° point of ESC Position Paper).

According to the guidelines, patients with a working diagnosis of STEMI do not require, and even it is not advisable, troponin assessment, should it delay revascularization (<120 min). Thus, the diagnosis of STEMI is sufficient with the EKG and clinical symptoms during first medical contact.

Ibanez B, James S, Agewall S, Antunes MJ, Bucciarelli-Ducci C, Bueno H, Caforio ALP, Crea F, Goudevenos JA, Halvorsen S, Hindricks G, Kastrati A, Lenzen MJ, Prescott E, Roffi M, Valgimigli M, Varenhorst C, Vranckx P, Widimský P; ESC Scientific Document Group. 2017 ESC Guidelines for the management of acute myocardial infarction in patients presenting with ST-segment elevation: The Task Force for the management of acute myocardial infarction in patients presenting with ST-segment elevation of the European Society of Cardiology (ESC). Eur Heart J. 2018; 39: 119-177.

2) There are no data on troponin. How was STEMI diagnosed?

According to the ESC guidelines, the diagnosis of STEMI can be made on the basis of the ECG itself and clinical symptoms during the first medical contact. 

Ibanez B, James S, Agewall S, Antunes MJ, Bucciarelli-Ducci C, Bueno H, Caforio ALP, Crea F, Goudevenos JA, Halvorsen S, Hindricks G, Kastrati A, Lenzen MJ, Prescott E, Roffi M, Valgimigli M, Varenhorst C, Vranckx P, Widimský P; ESC Scientific Document Group. 2017 ESC Guidelines for the management of acute myocardial infarction in patients presenting with ST-segment elevation: The Task Force for the management of acute myocardial infarction in patients presenting with ST-segment elevation of the European Society of Cardiology (ESC). Eur Heart J. 2018; 39: 119-177.

3) Table 1: The majority of MINOCA patients with prior AMI, had also prior PCI  so they supposed to have a significative CAD. How these patients could be classified as MINOCA?

Previous PCI does not exclude STEMI and MINOCA. STEMI MINOCA can be diagnosed both in patients following PCI and in those after CABG in the past - in other words, a positive history of advanced atherosclerosis does not exclude MINOCA.

4) Table 1: 67 patients classified as MINOCA had a previous CABG. The same as before, these patients had significative CAD. I suppose they are not MINOCA.

Here we have a similar justification as in the case of the point describede above.

5) Table 2: “Considering patients from the MINOCA group, 71.57% had significant stenoses while 28.43% no visible atherosclerosis”. If these patients had significant CAD, why they are called MINOCA?

In Table 2, it is shown that in the MINOCA group, 71.57% of patients had atherosclerotic lesions in their coronary arteries, but th significance was not confirmed. However, there is no information stating that 71.57% of the MINOCA group had significant stenosis.

6) What was the cause of MINOCA? Is there any information for the prevalence of plaque disruption, coronary artery spasm, coronary thromboembolism, coronary dissection?

Unfortunately, the data provided in the present publication come from a registry that collects only periprocedural data and is supplemented by operators of catheterisation laboratories. Therefore, we do not have data on the final etiology of MINOCA. It is widely known that to establish the etiology of MINOCA in a large percentage of cases, even many weeks after the initial incident, is conditioned by the need to perform additional tests, such as a thrombophilic test package, magnetic resonance imaging of the heart or other additional diagnostic tests. However, when it comes to finding coronary spasm or dissection, including spontaneous coronary dissection, we have such data, but due to the large percentage of missing data, we have refrained from including this in statistical calculations due to the high probability of bias and drawing erroneous conclusions. This was added to the “Strengths and limitations” section. 

7) It is sufficient to approximate the numbers in the tables to the first decimal. Tables will be more readable.

This was modified in Tables 1-3, however, we have left the data presented so far in Table 4-5, due to the need to maintain differences for individual confidence intervals.

8) 45% of MINOCA are in DAPT. In these patients, was DAPT therapy administered even without DES implantation?

The results of the analysed data, which are presented in the current publication, refer to the pre-hospital period, often directly pre-hospital, therefore, it is not possible to assess the impact on prognosis regarding the amount of antiplatelet drugs in the follow-up period. Certainly, in some patients, DAPT therapy is associated with recent PCI or acute coronary syndrome in recent months, but it is not possible to estimate the percentage based on the available data collected in the registry. This was clarified in the “Discussion section”. 

9) The score assigned to each variable is not clear.

The graph allows to show a scale for all main effects in the regression formula. Covariate distributions are superimposed on nomogram scales.

10) The main purpose is understanding how distinguish a priori a patient with MINOCA and possibly not to perform a coronary study. I believe that the results are absolutely not suitable for reaching this conclusion. Even if the score is > 650, the probability that it is a MINOCA is about 71%; therefore, it means that about 30% of patients are not MINOCA, actually; I don't think with these numbers it is possible to decide not to perform a CAG to a patient when it is recognized that an early PTCA can improve the prognosis. The criteria for performing a CAG as soon as possible (in STEMI) have been formulated precisely because they admit a low sensitivity and therefore provide for the possibility of having false positives with patients with free coronary arteries.

We definitely agree with the raised doubts, however, the fact that this approach is currently in place does not change the fact that some of the patients with STEMI do not have significant lesions in the coronary arteries, and in the future, it may be tempting to create such an algorithm that would with an almost 100% likelihood rule out patients with STEMI who do not require coronary angiography. Such possibilities can also be obtained from an accurate image scan that could be performed in fast mode. The purpose of this article is to spark a discussion on this topic. This was additionally clarified in the “Conclusions” section. 

11) AUC 0.7  It seems to me little bit low.

This is true, but please see the standard error of 0.000046, which is very small indicating very precise predictive value.

Reviewer #2: M. Jędrychowska and coworkers studied the main clinical and angiography differences in a large population of STEMI in both obstructive acute myocardial infarction and MINOCA patients. Data were collected between January 2014 and December 2019, and were selected from 1,177,218 patients who underwent coronary angiography. The authors aimed to create a tool facilitating a rapid separation of the MINOCA patients from the entire group of obstructive STEMI ones.

Finally, they evaluated a score able to address and separate MINOCA from obstructive AMI patients hospitalized due to MI with ST segment elevation. The model showed that patients who scored more than 600 points had a 19% probability of MINOCA, and for patients scoring more than 650 points, the probability of MINOCA was 71%. The other end of the MINOCA probability scale was marginal for patients who scored less than 500 points (< .2%).

The strengths of this study are:

- First, the large population of the study. Indeed they evaluated more than 1 million patients from the Polish National Registry of Percutaneous Coronary Interventions (ORPKI). They selected a broad sample of STEMI underwent PCI and MINOCA with ST-segment elevation (more than 5600 ones). Respect to the current literature on MINOCA, this is a very large population though from a retrospective registry.

- They collected only ST-segment acute myocardial infarction. This is a specific group of patients, especially in the MINOCA, which is worth studying due to higher mortality, particularly intra-hospital.

- The idea of a model able to distinguish STEMI obstructive AMI vs STEMI MINOCA could be very useful in clinical practice due to the subsequent diagnostic and therapeutic implications.

- I appreciate the table 2 in which they showed the coronary angiography and procedural indices.

However, there are some limitations of this study. The major pitfalls are:

- MINOCA patients are a heterogeneous entity. The current literature considered as MINOCA only coronary ischemic causes of acute myocardial infarction without obstructive coronary arteries. In our clinical practice, we have to exclude non-cardiac causes of troponin surge but also cardiac conditions like tako-tsubo syndrome or acute myocarditis.

In this work, the MINOCA population is quite heterogeneous and seemed quite unselected. They had not well-established inclusion and exclusion criteria for MINOCA diagnosis. Moreover, what were the main causes of MINOCA? Did they use cardiac magnetic resonance or other secondary diagnostic tools to address the final MINOCA diagnosis, as the recent guidelines recommend?

The current database covers only the pre-hospital period and is limited to the stay in at catheterisation unit, thus, we are not able to analyse the final diagnosis, due to the fact that diagnostic tests which are necessary to make a final diagnosis, such as cardiac magnetic resonance or examinations for thrombophilia, are performed after coronary angiography. The presented analysis includes patients prior to laboratory tests (cTn assessment), diagnosed with STEMI according to current ESC guidelines (based on ECG and clinical symptoms) during First Medical Contact (FMC). 

Ibanez B, James S, Agewall S, Antunes MJ, Bucciarelli-Ducci C, Bueno H, Caforio ALP, Crea F, Goudevenos JA, Halvorsen S, Hindricks G, Kastrati A, Lenzen MJ, Prescott E, Roffi M, Valgimigli M, Varenhorst C, Vranckx P, Widimský P; ESC Scientific Document Group. 2017 ESC Guidelines for the management of acute myocardial infarction in patients presenting with ST-segment elevation: The Task Force for the management of acute myocardial infarction in patients presenting with ST-segment elevation of the European Society of Cardiology (ESC). Eur Heart J. 2018; 39: 119-177.

- I think that the best clinical utility of the diagnostic model, created in this work, is to rule out the probability of MINOCA diagnosis. In fact, if the score is less than 500 points the probability of MINOCA is quite low < .2%. However, this model could not help the clinicians to avoid an urgent coronary angiography in patients with a suspected STEMI, even if the probability of obstructive CAD is low.

Yes, we definitely agree with this assumption, however the main aim of the current study was to raise a discussion, and in the future, make an attempt to create a scheme that reduces the number of unnecessarily performed invasive coronary angiography in patients with acute STEMI.

In the paper, the authors should explain more the clinical role of this predictive score (eg: in patients with high MINOCA probability, it’s useful to use more diagnostic tools, like left ventriculography during angiography or intravascular imaging, and to design a specific treatment in this population).

Thank you for that valuable remark. We added more information on this subject in the “Conclusions” section.

- It could be useful to create a predictive model with only pre-angiography variables, eg exclude P2Y12 inhibitors and add other variables like the presence of typical angina. This could be more useful for the clinicians as explained above.

Yes, we definitely agree, and thank you for this valuable remark, however the registry did not contain such information. As such, we are not able to include this data in the current analysis. 

Minor limitation:

- I suggest a more careful revision of the English language in the text

The text has been extensively and carefully proof-edited by a native speaker, professional translator and proofeditor (AmE Native Katarzyna Smith-Nowak, ul. Miejscowa 8, 30-499 Kraków, e-mail: english.native123@gmail.com, Phone: 0048 505 990 391, NIP: PL6792878140), to ensure proper cohesion, coherence and style of the manuscript, while maintaining its substantive content. The general readability for an English-speaking audience should be appropriate, the flow of the text ensured. 

Below, please find a point-by-point list regarding the types of alterations/items checked in the text: 

- Extensive stylistic editing 

- Eradication of repetitions and redundancies as well as spelling mistakes

- Correction of grammatical number 

- Correction of spacing, punctuation

- Exchanging colloquial linguistic items for more formal ones 

- Maintaining consistency of spelling 

- Correction of preposition usage 

- Correction of article usage 

- Maintaining linguistic consistency within the text

-Maintaining consistent usage of numerical format in data presentation 

- Corrections regarding parts of speech 

- Maintaining cohesion and coherence 

- Changes in word order 

We hope that the text in present form is in accordance with and meets all applicable standards of your renown Journal.

---

## [Decision Letter · Decision Letter 1]

11 May 2021

PONE-D-20-39776R1

ST-segment elevation myocardial infarction with non-obstructive coronary arteries: score derivation for prediction based on a large national registry

PLOS ONE

Dear Dr. Januszek,

Thank you for submitting your manuscript to PLOS ONE. After careful consideration, we feel that it has merit but does not fully meet PLOS ONE’s publication criteria as it currently stands. Therefore, we invite you to submit a revised version of the manuscript that addresses the points raised during the review process.

We look forward to receiving your revised manuscript.

Kind regards,

Raffaele Bugiardini, M.D.

Academic Editor

PLOS ONE

Reviewers' comments:

Reviewer's Responses to Questions

**Comments to the Author**

Reviewer #3: (No Response)

Reviewer #4: (No Response)

2. Is the manuscript technically sound, and do the data support the conclusions?

Reviewer #3: (No Response)

Reviewer #4: Partly

3. Has the statistical analysis been performed appropriately and rigorously? 

Reviewer #3: (No Response)

Reviewer #4: Yes

4. Have the authors made all data underlying the findings in their manuscript fully available?

Reviewer #3: (No Response)

Reviewer #4: Yes

5. Is the manuscript presented in an intelligible fashion and written in standard English?

Reviewer #3: (No Response)

Reviewer #4: Yes

6. Review Comments to the Author

Reviewer #3: This work intended to design a score for predicting MINOCA among STEMI patients in the Polish national registry of PCIs. They analyzed 124,663 pPCI treated individuals and 5,695 individuals with STEMI and MINOCA. The results showed the significant difference between patients with MINOCA and those in the MI-CAD group based on the proposed MINOCA score.

1. Please make the table self-explained. The notation needs to be noted. For example, 120 [60 ÷ 330]. What does it mean in bracket?

2. Need to precisely provide the information how the score was constructed in method section.

3. Validation is required to comment on the model performance. It’s unclear how the authors evaluate the performance of the score. It seems that the same sample was used. If so, it would lead to inflated results for the performance. If not, detail information should be provided.

Reviewer #4: This study has used a large national registry of cardiac interventional procedures to evaluate the STEMI population undergoing urgent angiography in relation the presence or absence of coronary artery disease (CAD). They compared the clinical features of STEMI patients with CAD (MICAD) to STEMI patients without CAD (MINOCA) and performed a regression model to identify predictors of MINOCA patients in this cohort.

This study is one of few that has focused on the STEMI population, which is an important knowledge gap, considering data which highlights MINOCA patients do not have a negligible risk of adverse outcomes, thus raising concerns about prognosis and appropriate management in the STEMI MINOCA population.

The authors developed a predictive score from their regression model (a nomogram) which showed the following factors were associated with MINOCA: younger age, female gender, no history of prior CABG, no history of smoking, no arterial hypertension, presence of COPD, no treatment with

3rd generation P2Y12, no direct transport to hospital, Killip class I or II, no history of diabetes and greater body mass at admission.

The authors present a very large population in the study and the clinical focus area is important, however there are some concerns with the methodology and the subsequent interpretation of these findings which at large do not serve the improved understanding of the MINOCA population.

The key concerns is in relation to the MINOCA definition and diagnosis.

The 4th Universal Definition of AMI states that MINOCA is:

MI patients with no angiographic obstructive CAD (> 50% diameter stenosis in a major epicardial vessel) AND, MINOCA, like the diagnosis of MI, indicates that there is an ischaemic mechanism responsible for the myocyte injury (i.e., non-ischaemic causes such as myocarditis have been

excluded).

There are challenges of the use or misuse of the term MINOCA since the term should be exclusively reserved for those patients with confirmed MI and thus who have undertaken key investigations such as cardiac MRI.

There is very limited data on such patients and institutions have limited availability of additional diagnostic testing. As such the presented cohort in this study, like many others, is a heterogenous cohort of ‘suspected MINOCA’. This should be acknowledged upfront.

The authors should adopt this language of ‘suspected MINOCA’ in their manuscript and refer to the 4th Universal Definition in their limitations.

The term ‘predictive score for MINOCA’ or ‘MINOCA predictive score’ should not be used as this is misleading due to the heterogeneity in the cohort.

The introduction should also be clarified as other non-coronary conditions such as myocarditis do not cause MINOCA but rather mimic MINOCA.

The scientific reasoning provided for the need for such a predictive score requires careful consideration also. The authors state that ‘coronary angiography is only an exclusive tool’ and whilst there is no relevance of primary PCI in MINOCA, the diagnostic angiogram in STEMI is still relevant to identify other underlying conditions such as Takotsubo or spontaneous coronary artery dissection.

Such a tool being used to ‘triage’ patients away from urgent angiography inappropriately may cause harm.

The aim in the introduction should be re-worded, in particular the phrase ‘reach a diagnosis’ should be removed.

The results show that the MINOCA patients are significantly younger compared to MICAD.

Where subsequent analyses age-adjusted to reflect this important difference, in particular, for CVD risk factors which are known to increase with ag?

In relation to coronary angiography data, the authors state 71.57% of MINOCA had significant stenoses. This is confusing.

Since MINOCA patients, by definition, have non-obstructive CAD, I assume the authors refer to plaques less than 50%?

Can the authors please clarify this and confirm that the MINOCA and MICAD populations were defined based on the 50% stenosis threshold. Were these thresholds ascertained by the angiography reports and/or operating physicians?

The nomogram description (“in the form of a nomogram, that is, a graphical representation of the relative impact of each prognostic factor within the global model”) in the results should be moved to the Methods.

In the conclusion, the main aims should be carefully re-considered

“The main aim of this publication was to spark a discussion, and in the future, try to create a

scheme reducing the number of unnecessarily performed invasive coronary angiographies in

patients with acute STEMI.”

It may be difficult to find clinical acceptance for a need to reduce unnecessary angiograms in STEMI.

However, the authors make an excellent point that a potential use of such predictive algorithms is for pre-paring the operator for a possible MINOCA diagnosis and thus for additional testing. This is a key point that should be further emphasised as a major issue in current MINOCA management is the lack of additional diagnostic testing. The discussion of a possible MINOCA diagnosis would also help prepare patients who are often left with no understanding of their presentation and discharged without a diagnosis.

Thus, the use of predictive algorithms in MINOCA has a place but the derivation population must be defined appropriately, and the focus of the tool is used to improve management and communication in the health care process.

7. PLOS authors have the option to publish the peer review history of their article (what does this mean?). If published, this will include your full peer review and any attached files.

Reviewer #3: No

Reviewer #4: No

---

## [Author Response · Author response to Decision Letter 1]

21 May 2021

Responses to the reviewers comments 

Reviewer #3: This work intended to design a score for predicting MINOCA among STEMI patients in the Polish national registry of PCIs. They analyzed 124,663 pPCI treated individuals and 5,695 individuals with STEMI and MINOCA. The results showed the significant difference between patients with MINOCA and those in the MI-CAD group based on the proposed MINOCA score.

1. Please make the table self-explained. The notation needs to be noted. For example, 120 [60 ÷ 330]. What does it mean in bracket?

This was corrected. 

2. Need to precisely provide the information how the score was constructed in method section.

3. Validation is required to comment on the model performance. It’s unclear how the authors evaluate the performance of the score. It seems that the same sample was used. If so, it would lead to inflated results for the performance. If not, detail information should be provided.

Response to comments 2 and 3. The ‘Methods’ section was improved, detailed methods for nomogram creation were presented. The bootstrap validation model was implemented, with the bias corrected index used to evaluate the model’s prognostic performance.

Reviewer #4: 

The key concerns is in relation to the MINOCA definition and diagnosis.

The 4th Universal Definition of AMI states that MINOCA is:

MI patients with no angiographic obstructive CAD (> 50% diameter stenosis in a major epicardial vessel) AND, MINOCA, like the diagnosis of MI, indicates that there is an ischaemic mechanism responsible for the myocyte injury (i.e., non-ischaemic causes such as myocarditis have been

excluded).

This was clarified in the ‘Methods’ and ‘Limitations” sections. 

There are challenges of the use or misuse of the term MINOCA since the term should be exclusively reserved for those patients with confirmed MI and thus who have undertaken key investigations such as cardiac MRI.

This was clarified in the ’Methods’ section and highlighted in the ‘Limitations’ section.

There is very limited data on such patients and institutions have limited availability of additional diagnostic testing. As such the presented cohort in this study, like many others, is a heterogenous cohort of ‘suspected MINOCA’. This should be acknowledged upfront.

This was modified in the ‘Methods’ section. 

The authors should adopt this language of ‘suspected MINOCA’ in their manuscript and refer to the 4th Universal Definition in their limitations. 

This was modified in the ‘Methods’ section.

The term ‘predictive score for MINOCA’ or ‘MINOCA predictive score’ should not be used as this is misleading due to the heterogeneity in the cohort.

This was modified according to the suggestions.

The introduction should also be clarified as other non-coronary conditions such as myocarditis do not cause MINOCA but rather mimic MINOCA.

This sentence has been rephrased, as recommended by the Reviewer. 

The scientific reasoning provided for the need for such a predictive score requires careful consideration also. The authors state that ‘coronary angiography is only an exclusive tool’ and whilst there is no relevance of primary PCI in MINOCA, the diagnostic angiogram in STEMI is still relevant to identify other underlying conditions such as Takotsubo or spontaneous coronary artery dissection.

Such a tool being used to ‘triage’ patients away from urgent angiography inappropriately may cause harm.

The aim in the introduction should be re-worded, in particular the phrase ‘reach a diagnosis’ should be removed.

This was corrected. 

The results show that the MINOCA patients are significantly younger compared to MICAD.

Where subsequent analyses age-adjusted to reflect this important difference, in particular, for CVD risk factors which are known to increase with ag?

The final multiple regression model that was used to create the nomogram includes age as a covariate, hence, the results are already age-adjusted.

In relation to coronary angiography data, the authors state 71.57% of MINOCA had significant stenoses. This is confusing.

This was corrected. 

Since MINOCA patients, by definition, have non-obstructive CAD, I assume the authors refer to plaques less than 50%?

This was clarified in the ‘Methods’ section. 

Can the authors please clarify this and confirm that the MINOCA and MICAD populations were defined based on the 50% stenosis threshold. Were these thresholds ascertained by the angiography reports and/or operating physicians?

This was modified. 

The nomogram description (“in the form of a nomogram, that is, a graphical representation of the relative impact of each prognostic factor within the global model”) in the results should be moved to the Methods.

This was corrected and this paragraph has been moved to the chapter on ‘Methods’.

In the conclusion, the main aims should be carefully re-considered

“The main aim of this publication was to spark a discussion, and in the future, try to create a

scheme reducing the number of unnecessarily performed invasive coronary angiographies in

patients with acute STEMI.”

It may be difficult to find clinical acceptance for a need to reduce unnecessary angiograms in STEMI.

However, the authors make an excellent point that a potential use of such predictive algorithms is for pre-paring the operator for a possible MINOCA diagnosis and thus for additional testing. This is a key point that should be further emphasised as a major issue in current MINOCA management is the lack of additional diagnostic testing. The discussion of a possible MINOCA diagnosis would also help prepare patients who are often left with no understanding of their presentation and discharged without a diagnosis.

Thus, the use of predictive algorithms in MINOCA has a place but the derivation population must be defined appropriately, and the focus of the tool is used to improve management and communication in the health care process. 

This was corrected.

---

## [Decision Letter · Decision Letter 2]

23 Jun 2021

PONE-D-20-39776R2

ST-segment elevation myocardial infarction with non-obstructive coronary arteries: score derivation for prediction based on a large national registry

PLOS ONE

Dear Dr. Januszek,

Thank you for submitting your manuscript to PLOS ONE. After careful consideration, we feel that it has merit but does not fully meet PLOS ONE’s publication criteria as it currently stands. Therefore, we invite you to submit a revised version of the manuscript that addresses the following points: 

We look forward to receiving your revised manuscript.

Kind regards,

Raffaele Bugiardini, M.D.

Academic Editor

PLOS ONE

Journal Requirements:

Additional Editor Comments:

Though review articles are appropriately used as overview citations for broad scientific topics or ideas, most citations, especially those focusing on previously published concepts or results, should be of original research papers. The community also values the accurate assignment of credit and precedence for scientific discoveries. As so, please include in your revision references of larger studies, reporting contemporary cohorts in order to inform the audience on the heterogeneity of patients with MINOCA, specifically:

Introduction, page  1, lines 2-3:  Circ Cardiovasc Qual Outcomes . 2017 Dec;10(12):e003443. doi: 10.1161/CIRCOUTCOMES.116.003443.Introduction page 1, lines 8-12: reference to this recently published original paper regarding the mechanism and etiologies of MINOCA should be added instead of the review paper-reference # 4: Circulation. 2021;143:624–640 doi: 10.1161/CIRCULATIONAHA.120.052008Introduction, page  2, lines 2-3:  Circ Cardiovasc Qual Outcomes . 2017 Dec;10(12):e003443. doi: 10.1161/CIRCOUTCOMES.116.003443 and Journal of the American Heart Association. 2020;9:e017235 doi: 10.1161/JAHA.120.017235Discussion in regard with the following statement” *What is interesting, in studies on the use of DAPT in MINOCA patients, it is shown that therapy is not only of no benefit, but may be detrimental in this group of patients [20,21]”*Reference # 20 could be misleading for the audience and should be revised accordingly. The study was an underpowered single-center study and as such is not able to detect the possible benefit or harm of therapeutic regimes in the MINOCA population. References of larger robust studies should be reported instead see Circulation.  2017 Apr 18;135(16):1481-1489. doi: 10.1161/CIRCULATIONAHA.116.026336*.*Reference # 21 refers to an abstract. The full manuscript recently published, should be reported instead: Heart. 2021 Jan 27;heartjnl-2020-318045. doi: 10.1136/heartjnl-2020-318045. PMID: 33504513. Additionally, with regard to the above-mentioned statement, it should be noted that, so far, possible harm associated with DAPT is mainly related to higher major bleeding rates, given also the higher incidence of MINOCA in women. In this regard the post-hoc analysis of the OASIS-7 trial (Heart. 2021 Jan 27;heartjnl-2020-318045. doi: 10.1136/heartjnl-2020-318045.) very nicely inform us about the possible harm related to intensified dosing strategy. Therefore, bleeding risk should be mentioned briefly.

Reviewers' comments:

Reviewer's Responses to Questions

**Comments to the Author**

1. If the authors have adequately addressed your comments raised in a previous round of review and you feel that this manuscript is now acceptable for publication, you may indicate that here to bypass the “Comments to the Author” section, enter your conflict of interest statement in the “Confidential to Editor” section, and submit your "Accept" recommendation.

Reviewer #3: All comments have been addressed

Reviewer #4: All comments have been addressed

2. Is the manuscript technically sound, and do the data support the conclusions?

Reviewer #3: (No Response)

Reviewer #4: Yes

3. Has the statistical analysis been performed appropriately and rigorously? 

Reviewer #3: (No Response)

Reviewer #4: Yes

4. Have the authors made all data underlying the findings in their manuscript fully available?

Reviewer #3: (No Response)

Reviewer #4: Yes

5. Is the manuscript presented in an intelligible fashion and written in standard English?

Reviewer #3: (No Response)

Reviewer #4: Yes

6. Review Comments to the Author

Reviewer #3: (No Response)

Reviewer #4: Dear Authors, comments have been addressed and incorporated into the manuscript.

Minor error on last page - MINOCA is spelt incorrectly "MINORC"

7. PLOS authors have the option to publish the peer review history of their article (what does this mean?). If published, this will include your full peer review and any attached files.

Reviewer #3: No

Reviewer #4: No

---

## [Author Response · Author response to Decision Letter 2]

23 Jun 2021

Responses to the reviewers comments 

PONE-D-20-39776R2

ST-segment elevation myocardial infarction with non-obstructive coronary arteries: score derivation for prediction based on a large national registry

PLOS ONE

Dear Dr. Januszek,

Thank you for submitting your manuscript to PLOS ONE. After careful consideration, we feel that it has merit but does not fully meet PLOS ONE’s publication criteria as it currently stands. Therefore, we invite you to submit a revised version of the manuscript that addresses the following points: 

• An unmarked version of your revised paper without tracked changes. You should upload this as a separate file labeled 'Manuscript'

Additional Editor Comments:

Though review articles are appropriately used as overview citations for broad scientific topics or ideas, most citations, especially those focusing on previously published concepts or results, should be of original research papers. The community also values the accurate assignment of credit and precedence for scientific discoveries. As so, please include in your revision references of larger studies, reporting contemporary cohorts in order to inform the audience on the heterogeneity of patients with MINOCA, specifically:

• Introduction, page 1, lines 2-3: Circ Cardiovasc Qual Outcomes . 2017 Dec;10(12):e003443. doi: 10.1161/CIRCOUTCOMES.116.003443.

This reference has been added. 

• Introduction page 1, lines 8-12: reference to this recently published original paper regarding the mechanism and etiologies of MINOCA should be added instead of the review paper-reference # 4: Circulation. 2021;143:624–640 doi: 10.1161/CIRCULATIONAHA.120.052008

The reference: Niccoli G, Scalone G, Crea F. Acute myocardial infarction with no obstructive coronary atherosclerosis: mechanisms and management. Eur Heart J. 2015; 36: 475-481. https://doi.org/10.1093/eurheartj/ehu469 PMID: 25526726 has been removed and instead of that suggested reference was added

• Introduction, page 2, lines 2-3: Circ Cardiovasc Qual Outcomes . 2017 Dec;10(12):e003443. doi: 10.1161/CIRCOUTCOMES.116.003443 and Journal of the American Heart Association. 2020;9:e017235 doi: 10.1161/JAHA.120.017235

This has been modified, and suggested referenses were addedd

• Discussion in regard with the following statement” What is interesting, in studies on the use of DAPT in MINOCA patients, it is shown that therapy is not only of no benefit, but may be detrimental in this group of patients [20,21]”

o Reference # 20 could be misleading for the audience and should be revised accordingly. The study was an underpowered single-center study and as such is not able to detect the possible benefit or harm of therapeutic regimes in the MINOCA population. References of larger robust studies should be reported instead see Circulation. 2017 Apr 18;135(16):1481-1489. doi: 10.1161/CIRCULATIONAHA.116.026336.

This reference (20) has been replaced by suggested citation. 

o Reference # 21 refers to an abstract. The full manuscript recently published, should be reported instead: Heart. 2021 Jan 27;heartjnl-2020-318045. doi: 10.1136/heartjnl-2020-318045. PMID: 33504513. 

This reference (21) has been replaced by suggested citation. 

o Additionally, with regard to the above-mentioned statement, it should be noted that, so far, possible harm associated with DAPT is mainly related to higher major bleeding rates, given also the higher incidence of MINOCA in women. In this regard the post-hoc analysis of the OASIS-7 trial (Heart. 2021 Jan 27;heartjnl-2020-318045. doi: 10.1136/heartjnl-2020-318045.) very nicely inform us about the possible harm related to intensified dosing strategy. Therefore, bleeding risk should be mentioned briefly.

This was corrected.

---

## [Editor Report · Decision Letter 3]

28 Jun 2021

ST-segment elevation myocardial infarction with non-obstructive coronary arteries: score derivation for prediction based on a large national registry

PONE-D-20-39776R3

Dear Dr. Januszek,

We’re pleased to inform you that your manuscript has been judged scientifically suitable for publication and will be formally accepted for publication once it meets all outstanding technical requirements.

Kind regards,

Raffaele Bugiardini, M.D.

Academic Editor

PLOS ONE
---

## [Editor Report · Acceptance letter]

27 Jul 2021

PONE-D-20-39776R3 

ST-segment elevation myocardial infarction with non-obstructive coronary arteries: score derivation for prediction based on a large national registry 

Dear Dr. Januszek:

I'm pleased to inform you that your manuscript has been deemed suitable for publication in PLOS ONE. Congratulations! Your manuscript is now with our production department. 

Kind regards, 

on behalf of

Prof. Raffaele Bugiardini 

Academic Editor

PLOS ONE